# Eco-Friendly and Economic, Adsorptive Removal of Cationic and Anionic Dyes by Bio-Based Karaya Gum—Chitosan Sponge

**DOI:** 10.3390/polym13020251

**Published:** 2021-01-13

**Authors:** Rohith K. Ramakrishnan, Vinod V. T. Padil, Stanisław Wacławek, Miroslav Černík, Rajender S. Varma

**Affiliations:** 1Institute for Nanomaterials, Advanced Technologies and Innovation (CXI), Technical University of Liberec (TUL), Studentská 1402/2, 461 17 Liberec, Czech Republic; rohith.kunjiparambil.ramakrishnan@tul.cz (R.K.R.); stanislaw.waclawek@tul.cz (S.W.); 2Regional Centre of Advanced Technologies and Materials, Palacký University in Olomouc, Šlechtitelů 27, 783 71 Olomouc, Czech Republic

**Keywords:** anionic-cationic dye removal, chitosan, conjugate sponge, karaya gum, sustainable materials, waste-water treatment

## Abstract

A novel, lightweight (8 mg/cm^3^), conjugate sponge of karaya gum (Kg) and chitosan (Ch) has been synthesized with very high porosity (~98%) and chemical stability, as a pH-responsive adsorbent material for the removal of anionic and cationic dyes from aqueous solutions. Experimental results showed that Kg-Ch conjugate sponge has good adsorption capacity for anionic dye methyl orange (MO: 32.81 mg/g) and cationic dye methylene blue (MB: 32.62 mg/g). The optimized Kg:Ch composition grants access to the free and pH-dependent ionizable functional groups on the surface of the sponge for the adsorption of dyes. The studies on the adsorption process as a function of pH, adsorbate concentration, adsorbent dose, and contact time indicated that the adsorption capacity of MB was decreased with increasing pH from 5 to 10 and external mass transfer together with intra-particle diffusion. The adsorption isotherm of the anionic dye MO was found to correlate with the Langmuir model (R^2^ = 0.99) while the adsorption of the cationic MB onto the sponge was better described by the Freundlich model (R^2^ = 0.99). Kinetic regression results specified that the adsorption kinetics were well represented by the pseudo-second-order model. The H-bonding, as well as electrostatic interaction between the polymers and the adsorption interactions of dyes onto Kg-Ch sponge from aqueous solutions, were investigated using attenuated total reflection-Fourier transform infrared (ATR-FTIR) spectroscopy, and the highly wrinkled porous morphology was visualized in depth by field-emission scanning electron microscopy (FE-SEM) analysis. Moreover, the samples could be reused without loss of contaminant removal capacity over six successive adsorption-desorption cycles. The hierarchical three-dimensional sponge-like structure of Kg has not been reported yet and this novel Kg-Ch sponge functions as a promising candidate for the uninterrupted application of organic pollutant removal from water.

## 1. Introduction

The extensive development in the industrial field, especially in textiles, cosmetics, leather, pharmaceutical, food, et cetera, gives rise to the production of large quantities of waste materials every day and most of them are released into water bodies [1,2,3]. These pollutants include dyes, oils, heavy metals, plastics, fertilizers, and many others. They are deleterious to the ecosystem as they cause toxic, carcinogenic, and mutagenic effects in microorganisms, other aquatic and terrestrial species, as well as human beings [4]. Among these contaminants, dyes—that provide colors to substrates—released into water create a serious threat to the environment due to their complex molecular structures, stability, and toxic intermediates produced as a result of partial degradation [5]. Thus, the elimination of these detrital substances from industrial effluents is one of the crucial challenges to chemists and environmentalists to sustain a greener environment.

Methods such as coagulation/flocculation [6], biological treatment [4], chemical oxidation [7], ozonation, chemical precipitation [7], photocatalytic degradation [8], membrane separation, electrochemical methodology, and adsorption [9] have been extensively applied to remove dye pollutants from wastewater. Among them, adsorption is regarded as the most competitive method because of its convenience, simplicity, economic feasibility, and efficiency [10]. Currently, several adsorbent materials, such as activated carbon, carbon-based nanomaterials, nanoporous alumina, mesoporous silica, zeolites, and hybrid xerogel have demonstrated their capability as effective adsorbents for dye contaminants [9,11,12]. However, their widespread use in water treatment is sometimes restricted due to the high cost and complexity of material preparation. In the past few years, numerous efforts have been focused on the development of more effective and economical adsorbents derived from natural biomass resources rather than commercial adsorbents. Bio-based materials such as cellulose [13], starch [14], lignin [15], alginate, and chitosan [16] have been used as dye adsorbents for many reasons. These include their affordability, low manufacturing costs, biocompatibility, biodegradability, hydrophilicity, copiousness, superior physicochemical properties, structural properties, and diversity of chemical functionalities [17]. Adsorption technology, a physical-chemical process, can benefit from the utilization of renewable biomaterials to interact with chemical species possessing high-performance attributes, on a par with synthetic materials [18]. The nature of the biomass-derived adsorbents that are highly beneficial in reducing further secondary contamination within the system is of utmost importance [19].

Three-dimensional hierarchical complex forms or sponges of natural polymers including tree gums give more stability and structural integrity in the water remediation field [20,21]. The presence of different active functional groups in the polymeric structure facilitates the complex formation and dye adsorption from water. Complex or conjugate sponges are formed by mixing various ratios of polymers and subsequently freeze-drying them under conditions that are favorable to gelation. Major considerations to dwell upon in the formation and stability of complexes are polymer concentration, ionic strength, pH of the reaction medium, nature and position of ionic groups on polymeric chains, nature of atomic and molecular interactions that can occur between different groups in polymer-polymer complexes, as well as the mixing ratios [21,22]. Many biopolymers have been utilized for the formation of sponges and complexes, the leading examples being chitosan, alginate, pectin, carrageenan, xanthan gum, chondroitin sulfate, gum Arabic, gum kondagogu, cellulose, and collagen [23,24,25].

A significant factor that relies on adsorption is the ability of the chemical functional groups on the adsorbent to interact with the dye molecules [26]. Dyes, based on their chemical structure and application, are classified into different types such as anionic, cationic, and non-ionic dyes. The cationic dyes such as methylene blue (MB), methyl violet (MV), et cetera, could be adsorbed by negatively charged materials while positively charged materials could uptake the anionic dyes such as acid red and methyl orange (MO) [27]. Most of the sponge adsorbents reported earlier could adsorb dye with a given charge type while sponges that could remove both anionic and cationic dyes in an economically feasible way are less explored. Conjugate sponges bearing both anionic and cationic groups would show pH-responsive charge type and strength, in other words, they will bring a negative charge or positive charge at different pH values, and thus perform as adsorbent of either cationic, anionic, or both, dyes.

Herein, we report a novel pH-responsive three-dimensional hierarchical sponge-like architecture of karaya gum (Kg) and chitosan (Ch) with properties of polyelectrolyte complex for adsorption of both anionic and cationic dyes from aqueous solution. Karaya gum is a high molecular weight acidic polysaccharide exudated from *Sterculia urens* [28] with a backbone consisting of α-d-galacturonic acid and α-l-rhamnose residues. Sidechains are attached by 1,2-linkage of β-d-galactose or by 1,3-linkage of β-d glucuronic acid to the galacturonic acid of the main chain. The high hydrophilicity and the lack of structural integrity in the three-dimensional structure prevent the use of Kg alone in water remediation or any other purposes. The conjugation of Kg with other natural polymers, for example, chitosan, reduces hydrophilicity, provides mechanical and chemical stability, and at the same time imparts plenty of diverse chemical functionalities. Chitosan is a biodegradable and biocompatible linear amino polysaccharide of β-d-glucosamine units joined by 1,4-linkages. The average linear charge density of chitosan is affected by the degree of deacetylation and by the solution pH since the former determines the density of the amino groups that can carry a charge while the latter signifies the degree of protonation of the amino groups. Using chitosan as a biopolymer, which is derived from chitin, has attracted researchers’ attention. The benefits of chitosan comprise its low cost, ease of polymerization and functionalization, and good stability [29]. The conjugation of Kg and Ch overcomes the limitations of either of the biopolymers alone and brings on the formation of a stable polyelectrolyte sponge having both negative and positive functional groups. At low pH, the amino groups in chitosan chains get protonated and the material exhibits a positive charge, while at high pH, the carboxyl groups of karaya gum get deprotonated which brings about a negative charge for the conjugate sponge. The electrostatic interactions between the opposite charges as well as the hydrogen-bonding interactions increase the mechanical as well as chemical stability of the conjugate sponge that contributes to its reusable nature without compromising contaminant removal efficiency. Mixing of polymers of opposite charge types in equal charge ratios creates a stoichiometric composition that precipitates out, while unequal mixing leads to a non-stoichiometric composition that enables plenty of functional groups available for the adsorption of substrates [30]. The Kg-Ch ratio has been optimized in the present work in order to obtain better adsorption performances. The physicochemical properties of the hydrogels were characterized, and the swelling profile was studied under different pH values. Methyl orange (MO) and methylene blue (MB) are well-known anionic and cationic dyes, which have been widely used in textile, printing, and research laboratories. Hence, MO and MB were selected as representative target pollutants in this study to investigate the pH-responsive selective adsorption capacities, adsorption mechanisms, and kinetics. The presence of abundant amino and carboxyl groups, higher porosity, and surface area coupled with pH-responsive nature would lead to desirable adsorption performances. The adsorption-desorption of dyes was also carried out for several cycles, to establish the reusability potential of the Kg-Ch sponge.

## 2. Materials and Methods

### 2.1. Reagents

Karaya gum (M_w_ = 1.5 × 10^6^ g/mol), low molecular weight chitosan (M_w_ = 9 × 10^4^ g/mol with a degree of deacetylation of 75–85%), acetic acid (CH_3_COOH), sodium hydroxide (NaOH), hydrochloric acid (HCl), methyl orange, and methylene blue were purchased from Sigma Aldrich (St. Louis, MO, USA). Acetone, chloroform, toluene, methanol, hexane, and tetrahydrofuran (THF) were obtained from Penta chemicals (Praha, Czech Republic). Ethanol and dimethyl sulfoxide (DMSO) were from Merck (Darmstadt, Germany). All the organic solvents are of analytical grade and used directly without further purification. Water was purified in the laboratory using a Milli-Q water system (Millipore, Billerica, MA, USA).

### 2.2. Preparation of Deacetylated Karaya Gum

Deacetylation of native karaya was done by following the method described by Vinod Vellora Thekkae Padil et al. [31] with slight modifications. Shortly, 1.0 *w*/*v*% dispersion of native karaya gum powder in deionized water was allowed to stand at constant temperature for 12 h to separate any undissolved matter, followed by filtration to obtain a clear solution. The clear gum solution was then deacetylated by mixing with 1.0 M NaOH with gentle agitation at room temperature and then pH was adjusted to 7.0 by the addition of 1.0 M HCl. The neutralized solution was finally freeze-dried to obtain the Kg powder.

### 2.3. Preparation of Chitosan Solution

Furthermore, 1.0 *w*/*v*% chitosan solution was prepared by dispersing the powder in deionized water and slowly adding 0.2 M acetic acid to render it completely soluble under acidic conditions; the pH of the final solution being 2.8.

### 2.4. Synthesis of Kg-Ch Sponge

Kg-Ch complexes were prepared by mixing Kg (1.0 *w*/*v*%) and Ch (1.0 *w*/*v*%) solutions in different Kg:Ch weight ratios (1:1, 1:2, 1:3, and 1:4) under 2000 rpm mechanical stirring at room temperature. Since all other combinations except 1:4, precipitated out, 1:4 composition was freeze-dried to obtain the Kg-Ch sponge. The electrostatic interaction between Kg and Ch is represented in Figure 1.

### 2.5. Zeta Potential

The zeta potentials of Kg, Ch, and Kg-Ch dispersions—of differing weight ratios—were measured at 25 °C using a ZetaPALS Potential Analyzer (Brookhaven, NY, USA) which determines zeta potential using phase analysis light scattering, a technique up to 1000 times more sensitive than traditional electrophoretic light scattering. The measurements were performed in triplicate for each sample and the average values reported.

### 2.6. Determination of Density

The density (ρ) of the sponge was calculated by measuring the weights and volumes of samples [32]. The weights (*M*) of the sponges were ascertained using an analytical balance while the volumes (*V*) of the sponges were determined using a digital micrometer at three different positions. The densities of the sponges were calculated according to Equation (1).
(1)$$\rho =\frac{M}{V}$$

### 2.7. Determination of Porosity

The porosity was measured gravimetrically using the ethanol displacement method [33] according to Equation (2),
(2)$$Porosity\;P\left(\%\right)=\frac{\left(m_{2}-m_{1}\right)\rho_{1}}{m_{1}\rho_{2}}\times 100$$
where *m*_1_ is the weight of the dry sample (g); *m*_2_ is the weight of sample with adsorbed ethanol (g); ρ_1_ is the density of sponge (g/mL), and ρ_2_ is the density of ethanol (g/mL). Three parallel samples were utilized and average values are reported.

### 2.8. Swelling Measurements

Evaluation of the swelling index (SI) of the sponge was performed by immersing 0.1 g of each sample in 100 mL distilled water at different pH values (adjusted with 0.1 M HCl/NaOH). The weighed, dried sponges were incubated at 25 °C for 24 h (under various pH conditions ranging from 2 to 12). The swollen adsorbents were removed from the water and the surface blot dried using filter paper. The swelling index was then calculated using Equation (3),
(3)$$SI=\frac{W_{S}-W_{D}}{W_{D}}\times 100$$
where *W_S_* and *W_D_* are the weights of samples after being swollen in water and the dry state, respectively. All swelling experiments were performed in triplicate and the mean values of data obtained were plotted.

### 2.9. Stability Studies

The chemical stability of the Kg-Ch sponge was tested by immersing the samples for 24 h under constant shaking of 300 rpm in common organic solvents (methanol, ethanol, toluene, acetone, DMSO, THF, chloroform, and hexane) and aqueous solution. Generally, the diffusion of the solution and solvents into the porous structure of the sponges is slow, due to the entrapped air within the structure that must be replaced by the solvents.

### 2.10. ATR-FTIR (Attenuated Total Reflection-Fourier Transform Infrared) Spectroscopy

An ATR-FTIR (Nicolet iZ10, Thermo Scientific, Waltham, MA, USA) spectrometer equipped with fixed 45° angle and horizontal ATR accessory with diamond crystal was used to characterize the functional groups present in native Kg, deacetylated Kg, Ch, and Kg-Ch sponge.

### 2.11. FE-SEM (Scanning Electron Microscopy)

Field-emission scanning electron microscopy (FE-SEM) characterization of the sponge was performed by a ZEISS, SIGMA (Oberkochen, Germany) at an accelerating voltage of 10 kV. Samples were stuck on the sample holder with a carbon pad and sputter-coated with gold. The average pore diameter was assessed using Image J software.

### 2.12. Thermogravimetric Analysis (TGA)

Thermal stability and composition were determined using Q500 Thermogravimetric Analyzer (TA Instruments, New Castle, USA). The analysis was carried out in a nitrogen atmosphere at a flow rate of 60 mL/min and the sample was heated from 30 to 600 °C at a heating rate of 10 °C/min.

### 2.13. Adsorption Studies of Dye Solutions

Methylene blue (MB) and methyl orange (MO) solutions were used in the dye adsorption tests. A stock solution was prepared at a concentration of 100 mg/L and diluted to 10, 20, 30, 40, and 50 mg/L. A 15 mL volume of the dye solution was employed in the adsorption experiments with a 0.02 g portion of the adsorbent whilst stirring on a magnetic stirrer bar rotating at 300 rpm. The amount of dye adsorbed by the adsorbent was determined by measuring the dye concentration in solution before and after the adsorption experiment using UV-Visible spectroscopy. The equilibrium adsorption amount *q_e_* (mg/g) and removal efficiency or removal capacity *R* (%) were calculated using Equations (4) and (5), respectively.
(4)$$Adsorption\;equilibrium,\;q_{e}=\frac{\left(Co-Ce\right)}{m}V$$
(5)$$Removal\;capacity,\;R\%=\frac{\left(Co-Ce\right)}{Co}100$$ where, *q_e_* is the amount of dye adsorbed at equilibrium, *C*_0_ is the initial dye concentration (mg/L), *C_e_* is the equilibrium dye concentration (mg/L), *m* is the weight of adsorbent (g) and *V* is the volume of the dye solution (L).

### 2.14. Adsorption Isotherm

The adsorption isotherm describes the relationship between the amount of dye adsorbed by the adsorbent and the concentration of dye remaining in the solution. Based on the adsorption isotherm at 30 °C, the adsorption mechanism was studied using the Langmuir and Freundlich models which can be expressed as follows (Equations (6) and (7)):(6)$$\frac{C_{e}}{q_{e}}=\frac{1}{q_{0}\;.\;\;K_{L}}+\frac{1}{q_{0}}C_{e}$$
(7)$$\log q_{e}=\log K_{F}+\frac{1}{n}\log C_{e}$$
where *C*_e_ (mg/L) is the equilibrium dye concentration and *q*_e_ (mg/g) is the equilibrium adsorption capacity. *K_L_* (L/mg), *K_F_* (L/mg) *q*_0_ (mg/g), and *n* represent the Langmuir constant, Freundlich constant, maximum adsorption capacity, and adsorption intensity, respectively.

### 2.15. Kinetic Studies

Adsorption kinetic models apply to the interpretation of adsorption data to gain an insight into adsorption efficiency, rate, and the rate-controlling step. To study the adsorption kinetics of Kg-Ch sponges, the adsorption capacities against the contact time of sponges in MB and MO solutions (with concentrations of 10, 20, 30, 40, and 50 mg/L) were investigated at 30 °C. Two well-known adsorption models, a pseudo-first-order model, and a pseudo-second-order model were used throughout to investigate the adsorption mechanism and kinetics of the sponges towards MB and MO. Equations (8) and (9) define the models as follows:(8)$$\mathrm{Pseudo}-\mathrm{first}-\mathrm{order}\;\mathrm{model},\log\left(q_{e}-q_{t}\right)=\log q_{e}+\frac{K_{1}}{2.303}\times t$$
(9)$$\mathrm{Pseudo}-\mathrm{second}-\mathrm{order}\;\mathrm{model},\;\frac{t}{q_{t}}=\frac{1}{K_{2}q_{e}^{2}}\frac{t}{q_{e}}$$
where, *q_e_* (mg/g) is the amount of dye adsorbed at equilibrium, *q_t_* (mg/g) is the adsorption capacity at a certain contact time *t* (min); *K*_1_ (L/min) and *K*_2_ (g/mg·min) are the rate constants of pseudo-first-order and pseudo-second-order models, respectively.

### 2.16. Regeneration and Reusability

The adsorption-desorption of the MB and MO dye molecules from the Kg-Ch sponge and the regeneration of the adsorbent was studied for six successive cycles of adsorption and desorption. Desorption of the MB and MO dye molecules was performed in 0.1 M HCl and 0.1 M NaOH, respectively, and the desorbed sponges were washed with distilled water. The regenerated sponges were then repeated for six successive cycles of dye adsorption. The dye removal rate was calculated using Equation (10),
(10)$$\mathrm{Removal}\;\mathrm{of}\;\mathrm{dye}\;\left(\%\right)=\;\frac{C_{0}-C_{t}}{C_{0}}100$$
where *C*_0_ is the initial concentration of dye and *C_t_* is the dye concentration at time *t*.

## 3. Results and Discussion

### 3.1. Kg-Ch Sponge

In order to investigate the effects of weight ratios in the properties of the conjugates, the physical appearances and zeta potentials of the Kg-Ch solutions were noted (see Appendix A). The 1:4 w% ratio of Kg-Ch was ascertained to be the optimum ratio to generate sponges. It was observed that during the preparation of the various Kg-Ch weight combinations (1:1–1:4), except for the 1:4 ratio, all other combinations resulted in precipitate formation. Mixing anionic and cationic polymers results in the formation of a polyelectrolyte complex with reversible electrostatic linkages due to their spontaneous association. If they combine in such a ratio that there is an excess of one charge, non-stoichiometric complex ensues which is soluble in the reaction medium while mixing of oppositely charged polyelectrolyte in equal charge ratio proceeds to the formation of stoichiometric polyelectrolyte complex which is insoluble and precipitated out. Zeta potential of pure Ch is 78.5 mV and that of Kg is −21.0 mV which is due to the presence of amino groups in Ch and carboxylic groups in Kg, giving them positive and negative net charges, respectively. The charge of Kg-Ch combinations is lower than that of Ch, indicating an interaction between the positively charged Ch and negatively charged Kg, which results in the neutralization of the free positive charge associated with ionized Ch upon the addition of Kg. At a 1:4 ratio, the conjugate has a zeta potential value of 65.8 mV and thus more amino-functional groups are accessible that could bind with dyes. Thus, a 1:4 Kg-Ch combination was selected for the synthesis of a conjugate sponge. This clear solution upon freeze-drying leads to the formation of a three-dimensional hierarchical structure with uniformly distributed pores for high dye adsorption.

### 3.2. Density and Porosity 

Density and porosity are important parameters that determine the quality and utility of adsorbent material. The density of the Kg-Ch sponge is 8.0 ± 0.3 mg/cm^3^ and porosity is ~98%. The very high porosity is due to the hierarchical microporous structure of the sponge, formed as a consequence of water crystals being evaporated during the freeze-drying process.

### 3.3. ATR-FTIR Analysis

The spectrum of Ch (Figure 2c) displayed two amine stretching peaks, ~3400 and ~3300 cm^–1^, covered by the broad peak of O–H stretching and the NH_2_ bending peaks of primary amine (position is dependent on surrounding structure); a CH_2_ bending peak and C–O–C stretching peaks at 1650, 1596, 1416, and 1030 cm^–1^, respectively. The peak appearing at 1374 cm^−1^ is the result of C–H stretching deformations. The FTIR spectra of the Kg-Ch sponge (Figure 2d) showed a shift in the NH_2_ bending peaks to lower wavenumber (1630, 1589 cm^–1^) indicating an electrostatic interaction between the protonated amine groups of Ch and the COO^−^group of Kg [34]. The stretching amine peaks and CH_2_ bending peak of Ch were also shifted because the COO^−^ in Kg could form intermolecular hydrogen bonds with Ch. This suggests that the Kg-Ch sponge composed of Ch and Kg is indeed stabilized through electrostatic interactions and intermolecular hydrogen bonding. Furthermore, the appearance of a broad absorption band (between 3600 and 3000 cm^−1^) due to O–H and N–H stretching and broadening of the amine bands (at 1589 cm^−1^) and carboxylate peak (at 1421 cm^−1^) also suggests the interactions between Kg and Ch occur employing either hydrogen or ionic bonding [35].

### 3.4. Chemical Stability 

The application of sponges as adsorbents requires chemical stability under different environments, both aqueous and organic. To evaluate the chemical stability of the Kg-Ch sponge, it was immersed in methanol, ethanol, toluene, acetone, DMSO, THF, chloroform, hexane, and deionized water—all in closed bottles at 25 °C for 24 h (Appendix A). The sponges were found to be highly stable after their immersion in various organic and aqueous solvents, reflecting the amphiphilic character of the Kg-Ch sponges and the utility of the same under different environments for contaminant removal. The chemical stability of the sponge has ensued from the high interaction between charged polymer components (Kg and Ch), both H-bonding and electrostatic.

### 3.5. SEM Analysis

The FE-SEM images of the Kg-Ch sponge is presented in Figure 3. It is highly porous with ordered pores, having an average size of about 57 µm (measured using ImageJ software). The wrinkled morphology of the pore walls, as clearly seen from the high-resolution image (Figure 3c), provides high surface area and more adsorption sites for the dye molecules that greatly influence its adsorption performance. 

### 3.6. Thermogravimetric Analysis

Figure 4 shows the thermograms of Ch, Kg, and Kg-Ch sponge with the derivative thermogravimetric curve insert. It was noted that the maximum degradation of Ch and Kg was observed at around 300 and 287 °C, respectively, while that of the Kg-Ch sponge at 273 °C. It can be considered as evidence of Kg-Ch complexation [36]. The shift to a lower degradation temperature in the Kg-Ch sponge indicates that the formation of ionic bonds between Ch and Kg probably correlates with a loss of organization. This occurs in two stages, the first being evaporation of water which transpires in the 50–100 °C temperature range—resulting in weight losses of 13.42%, 9.22%, and 7.36% (for Ch, Kg, and Kg-Ch), respectively. The individual polymers possess a high affinity towards water compared to the conjugate, the weight loss from the latter (in the first stage) emanating from water trapped in the conjugate during Kg-Ch sponge formation. In the second stage, the thermal degradation temperatures of Kg and Ch were 287 °C and 300 °C, respectively, corresponding to weight losses of 71% and 66%. Thermal degradation of the biopolymers is due to the depolymerization of their chemical networks. The degradation temperature of the Kg-Ch sponge occurs at 273 °C, accompanied by a 72.19% weight loss. The fact that the thermal decomposition temperature of the Kg-Ch sponge is lower than that observed for pure biopolymers is indicative of two phenomena, namely, the formation of ionic bonds between Ch and Kg and the loss of organizational structure.

### 3.7. Swelling Studies 

The ionic interactions between positively charged Ch and negatively charged Kg display pH-sensitive swelling. The charge balance inside the gelling network changes as the pH of the swelling medium varies (ranging from 2.0 to 12.0).

In an acidic medium, free positive charges (NH_3_^+^) appear due to the protonated amino groups in Ch and at the same time, Kg becomes neutralized within the gel. In contrast, when the medium is basic, Ch is neutralized and free negative charges (COO^−^) appear inside the gel as a result of the deprotonation of carboxylic groups on Kg. The mutual repulsion between positive and negative charges, coupled with the entry of water causes swelling. Uptake of water is also associated with the ability of unionized groups present in the polymers to form ionic bonds with water. Figure 5 shows the swelling index of Kg-Ch sponge under different pH conditions. At pH 6–10, the sponge shows a comparatively lower degree of swelling. This difference is due to the difference in ionization of the two polymers at different pH. The alkaline pH up to 10 is not as effective as pH 12 to ionize the carboxyl groups in Kg and similarly with the case of amino groups at pH 6. Since the free amino groups are high (as known from the zeta potential analysis) in Kg-Ch sponge, it facilitates the highest swelling, particularly at acidic pHs because of the formation of the high number of free positive charges.

### 3.8. Adsorption Properties of Kg-Ch Sponges

#### 3.8.1. Effect of pH 

Studies on the swelling index of Kg-Ch sponge in different pH conditions have already shown that the charge type of the sponge could be adjusted by the external pH values. Adsorbents with electric charges could effectively remove charged pollutants, such as cationic and anionic dyes. Thus, by responding to the pH values, the adsorption capacities of the Kg-Ch sponge for cationic and anionic dyes could be controlled. As depicted in Figure 6, the adsorption capacities of the dyes were seen to be highly dependent on the pH of the solution.

The pH of the solution activates the functional groups of adsorbents to impart a specific surface charge and also promoting dye speciation by affecting the degree of ionization of the dye during adsorption [37]. At low pH conditions, the amino groups of the Kg-Ch sponge get protonated to bring out a positive charge to the adsorbent which facilitates the electrostatic interaction between the sponge and anionic dye MO [38]. The maximum removal efficiency of MO by the sponge was found to be 14.3 mg/g (at *C*_0_ = 20 mg/L; contact time: 120 min; a mass of Kg-Ch = 20 mg; temperature: 25 °C) under acidic conditions that remains almost the same for the pH ranging from 2 to 6. At high pH, the deprotonated carboxyl groups furnish the sponge with negative charges which in turn enhances the interaction of the sponge with the cationic dye MB. Thus, the adsorption capacities of the Kg-Ch sponge for MB increased drastically with the increase of pH, reaching a maximum (14.5 mg/g) at a pH of 8.

From the above results, it follows that the Kg-Ch sponge can act as a highly efficient adsorbent for the adsorption of both MO and MB. Since the highest adsorption for MO and MB was noted at pH 4 and 8, respectively, the same pH was adopted in further experimentations.

#### 3.8.2. Effect of Adsorbent Dosage

The adsorption capacities of MO and MB dyes at different dosages of the Kg-Ch sponge are shown in Figure 7. The adsorbed amounts of dyes have certainly risen with the increasing weight of the sponge sample owing to the increased adsorbent surface area and the availability of more adsorption sites. However, the adsorption capacity (q_e_) presented the opposite trend with an increase in adsorbent concentration [39]. At higher sample (sponge) to solute concentration ratios, there is a very fast superficial sorption onto the adsorbent surface that lowers the solute concentration in the solution than when the sponge to solute concentration ratio is lower. The adsorption capacity, q_e_ (mg/g) was found to have decreased with increasing adsorbent mass due to the split in the flux or the concentration gradient between solute concentration in the solution and the solute concentration on the surface of the adsorbent [40]. Thus, with increasing adsorbent mass, the amount of dye adsorbed onto the unit weight of adsorbent gets reduced by the overlapping of adsorption sites in the adsorbent [41,42].

The adsorption capacity of MB decreased from 47.75 to 10.94 mg/g with increasing adsorbent concentration from 5 to 25 mg. This was attributed to two factors—the adsorption competition among adsorbates and the split in the concentration gradient. It was also discovered that the increase in adsorbent dosage (from 5 to 25 mg) resulted in a decrease in the amount of adsorbed MO dye, from 55.94 to 11.62 mg/g. From the results, it is apparent that the quantity of adsorbent and the adsorbed amount per unit weight are inversely proportional to each other.

#### 3.8.3. Effect of the Initial Concentration of Adsorbates

Dye adsorption experiments were carried out using varying dye concentrations ranging from 10 to 50 mg/L. The effects of initial concentration on the adsorption capacity of dyes are depicted in Figure 8. It was established that the equilibrium adsorption (q_e_) capacities of MB and MO dyes increased from 6.85 to 32.62 mg/g and 7.13 to 32.81 mg/g, respectively, with the initial dye concentration increasing from 10 to 50 mg/L. When the initial concentration of dyes increased, the driving force for mass transfer became larger and the interaction between the dyes and sponge (the adsorbent) was enhanced. This in turn resulted in a higher adsorption capacity of the sponge [43].

### 3.9. Adsorption Isotherm Evaluation

The adsorption isotherms were evaluated at 30 °C. The equilibrium adsorption capacities of MO and MB as a function of contact time are plotted in Figure 9a,b, the adsorption capability being dependent on the initial concentrations of the dye solutions. In the case of MO, the first stage of dye adsorption is quick and as time goes on, a gradual decrease is observed to attain equilibrium. This rapid initial stage is more pronounced at higher concentrations for MB. This behavior is related to the fact that, at the initial stage, a large number of vacant adsorption sites are available for adsorption, and later with increasing contact time, the remaining vacant adsorption sites are difficult to occupy because of repulsive forces between the dye molecules on the adsorbent and in the solution. The *C*_e_ (equilibrium dye concentrations) were recorded at the adsorption equilibrium with different *C_0_* (initial dye concentration) values of MB and MO solutions (10, 20, 30, 40, and 50 mg/L). As the value of *C_0_* increased, the Kg-Ch sponge exhibited higher adsorption capability. This was as high as 32.62 mg/g at the *C_e_* of 6.49 mg/L (*C_0_* = 50 mg/L) and 32.81 mg/g at the *C_e_* of 6.24 mg/L (*C_0_* = 50 mg/L) for MB and MO dyes, respectively.

The Langmuir and Freundlich adsorption isotherm models are employed for the prediction of the adsorption of MB and MO onto Kg-Ch sponge systems. According to the Langmuir isotherm model, maximum adsorption corresponds to the formation of a saturated monolayer of the adsorbate molecules on the adsorbent surface. The fitted equilibrium data for dye adsorption of MB and MO using the Langmuir and Freundlich isotherm models are plotted as shown in the Appendix A. All the equilibrium parameters obtained according to the Langmuir and Freundlich models are listed in Table 1, the Langmuir model showing a linear relationship, the correlation coefficients (R^2^) being 0.91 and 0.99 for MB and MO, respectively. Based on the Langmuir isotherm, the maximum adsorption capacities of the dyes MB and MO were 80.58 and 37.24, respectively; the Langmuir isotherm showing a better fit for MO. The fitting result presented that the adsorption behavior occurred on homogeneous surfaces of Kg-Ch sponge in a monolayer manner. Moreover, according to the Langmuir fitting result, the maximum adsorption capacity of the sponge for MO is 37.24 mg/g, which is in concordance with the experimental equilibrium value (32.8 mg/g).

The Freundlich model suggests a non-uniform affinity for adsorption onto heterogeneous surfaces. This non-uniformity is attributed to the presence of different functional groups in the Kg-Ch sponge. The K_F_ and n values were calculated from the plot of log *q_e_* against log *C_e_*, which also showed a linear relationship with the R^2^ values of 0.99 and 0.87, for MB and MO, respectively. Based on the R^2^ values, the adsorption of MB can be well described by the Freundlich isotherm model.

The adsorption capacity of the Kg-Ch sponge for dye (MB and MO) removal is comparable to that of the majority of natural and synthetic adsorbents reported in the literature [44,45,46,47,48,49]. It is therefore noteworthy that the Kg-Ch sponge has substantial potential for the removal of anionic and cationic dyes from aqueous solutions.

### 3.10. Adsorption Kinetics 

The adsorption depends on the characteristics of the adsorbent and the mass transfer process. To investigate the adsorption mechanism of MB and MO adsorbed onto the Kg-Ch sponge, two kinetic models have been used, namely, pseudo-first-order and pseudo-second-order.

The pseudo-first-order models for MB and MO are approximately linearly plotted in Figure 10a,b, respectively, and the pseudo-first-order constants and equilibrium adsorption densities were determined from the slopes and intercepts of the plots. Experimental data such as adsorption capacities (q_e_) are considerably different from the theoretical data. Table 2 illustrates that the variations in the correlation coefficients (R^2^) for the pseudo-first-order kinetic model suggest that the adsorptions (of both dyes) do not follow pseudo-first-order reaction kinetics.

The pseudo-second-order kinetic data for MB and MO was derived from the linear curves of Figure 11a,b, respectively. The pseudo-second-order constants and equilibrium adsorption capacities were determined from the slopes and intercepts of the linear plots, as listed in Table 3. The straight-line curves and correlation coefficients (R^2^ > 0.99) are indications of good agreement between experimental data and the pseudo-second-order kinetic model, for both MO and MB dye adsorption. The adsorption is controlled by outer diffusion and inner diffusion followed by the very fast adsorption of the adsorbate to the active sites on the adsorbent. In outer diffusion, the dye molecules move across the liquid film to the exterior surface of the sponge, and in inner diffusion, the dye molecules transport from the exterior surface to the interior of the sponge through the porous structure. 

The various natural adsorbents used for the removal of dyes (both MO and MB) with their uptake capacity are presented in Table 4. The values of adsorption capacities (q_e_) in the table are based on the Langmuir adsorption isotherm. The biosorption capacity of the Kg-Ch sponge for dye (methylene blue and methyl orange) removal is comparable to that of the majority of natural and synthetic adsorbents reported in the literature (Table 4). It is therefore noteworthy that the Kg-Ch sponge has substantial potential for the removal of dyes from aqueous solutions.

### 3.11. The Mechanism for the Adsorption of Dyes onto the Kg-Ch Sponge

The possible mechanisms for the adsorption of dye molecules onto Kg-Ch sponge are schematically illustrated in Figure 12. The polyionic charged surface of the Kg-Ch sponge could provide adsorption sites for electrostatic interactions with both MB and MO. It possesses a regular hierarchical structure with very high porosity, which can promote extensive contact between dye molecules and the sponge structure.

The ATR-FTIR spectra of the Kg-Ch sponge (before and after MB and MO adsorption) were compared to shed more light on the chemical bonding between the adsorbate and the adsorbent (Appendix A). It was observed that the spectra of Kg-Ch (after the adsorption of MB and MO) were different when compared to that seen for the sponge structure on its own, before dye adsorption. Under acidic conditions, the structure of the Kg-Ch sponge was found to be amenable towards adsorbing MO molecules, owing to the fact that, the protonated (−NH_3_^+^) form of the amino groups on chitosan can easily attract the −SO_3_^−^ groups on the MO molecule. After MO adsorption, the peaks noted in the ranges 3000–2800 cm^−1^ and 900–550 cm^−1^ corresponded to the C-H stretching vibrations of the benzene ring. The peaks at 1609 and 1523 cm^−1^ were ascribed to the C=C stretching vibrations in benzene rings while the adsorption peaks at 1313 cm^−1^ were assigned to S=O stretching vibrations from sulfonyl groups. This indicates that the MO dye was adsorbed effectively onto the Kg-Ch sponge, which is essentially accomplished by the −SO_3_Na group (on the dye) being adsorbed onto the adjacent −NH_2_ group (on the Ch) [59]. Furthermore, the structure of the sponge, together with its high porosity, provided more adsorption sites. Besides, the MO molecule has a linear structure with negative groups that favor strong electrostatic attraction towards the cationic species (−NH_3_^+^) present on Ch, thereby contributing to the highly selective loading of MO onto the Kg-Ch sponge.

The presence of Kg in the sponge provides more functional groups (such as carboxylic and hydroxyl) that can interact with MB dyes through the joint phenomena of electrostatic interactions and hydrogen bonding. FTIR spectra of the sponge after MB adsorption exhibited peaks at 2872 and 2927 cm^–1^ which were attributed to the stretching vibrations of aromatic –CH and methyl (–CH_3_) groups on MB [60]. There were also new peaks pertaining to the absorption peaks of MB (at 1603 and 1384 cm^−1^) which corresponded to aromatic ring vibrations and symmetric bending vibrations of the methyl (-CH_3_) group. This unambiguously confirms that interaction had indeed occurred between MB and the Kg-Ch sponge. In alkaline conditions, the deprotonated medium augmented the possible electrostatic interactions between COO^–^ groups of the adsorbent and the positively charged active sites of MB dye molecules.

### 3.12. Effect of Adsorption-Desorption Cycles on the Dye Removal Efficiency of Kg-Ch Sponge

The adsorption-desorption experiments for evaluating the regeneration and reusability potentials of the Kg-Ch sponge were conducted using 0.1 M NaOH (for MO) and 0.1 M HCl (for MB) in order to monitor dye adsorption (for six successive cycles) onto the sponge. The Kg-Ch sponge was utilized to adsorb both dyes for the same number of cycles and the removal efficiency was calculated concerning the first cycle (the results are presented in Figure 13). The adsorption capacity for MB was observed to decrease marginally during the repeated cycles while the MO dye adsorption capacity remained more or less constant throughout all the cycles. After a sequence spanning six cycles, the adsorption capacity of the sponge remained at 94% (for MO) and 90% (for MB)—in comparison with the first cycle—vividly demonstrating the reliable recyclability potential of the Kg-Ch sponge for consecutive operations. The structural integrity of the material reported in the present case was not been altered considerably during the successive absorption desorption cycles. The dye absorption ability of >90% even after the sixth cycle authenticates this point. However, a slight degree of deformation occurred as a result of force applied to remove the absorbed dye for repeating the cycle. Some of the pores might have collapsed and thus the material shows slightly decreased absorption capacity at the end of the sixth cycle. It has been reported earlier that the squeezing force should not be too high to sustain the structure of the sponge type materials [61]. If the squeezing force becomes high, it leads to the destruction of some of the pores and subsequent morphological change.

## 4. Conclusions

We have developed a novel eco-friendly sponge-like, pH-responsive adsorbent material based on gum karaya and chitosan for the efficient removal of anionic dye methyl orange and cationic dye methylene blue from aqueous solutions. High porosity (~98%) and chemical stability in a wide range of solvents together contributed to the high contaminant removal efficiency and reusability potential of the Kg-Ch sponge. It registered excellent adsorption of 32.8 mg/g for MO and 32.6 mg/g for MB at pH 4 and pH 8, respectively, due to the selective charge development in the sponge in response to the pH of the solution. The optimized composition of Kg and Ch not only benefited in the optimum interaction between the polymer components to enable chemical and mechanical stability but also the accessibility of specific functional groups-both free and ionizable- for the adsorption of dyes. The amount of dye uptake was found to increase with the increase in dye concentration and contact time and found to decrease with an increase in adsorbent dose. The adsorption process was controlled jointly by the mass transfer and the intra-particle diffusion. The Langmuir model is fitted for the adsorption isotherm of MO while the adsorption isotherm of MB is very well explained by the Freundlich model. A pseudo-second-order model was deemed to be suitable for interpreting the adsorption kinetics of both dyes (MB and MO) onto the sponge. The sponge system presented in this study can realistically be considered to be an efficient adsorption medium for both cationic and anionic dye removal having good reusability potential without compromising the adsorption capacity. It could well inspire further studies that can lead to improvements in selective adsorption and separation capabilities towards different organic, inorganic, and metal pollutants from aqueous media.

## Figures and Tables

**Figure 1 polymers-13-00251-f001:**
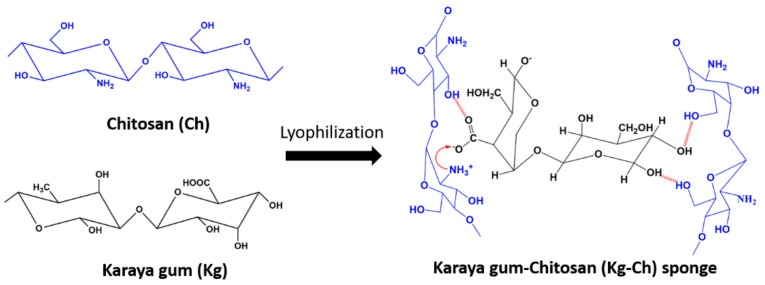
Schematic representation of the formation of Kg-Ch sponge from Karaya gum and chitosan.

**Figure 2 polymers-13-00251-f002:**
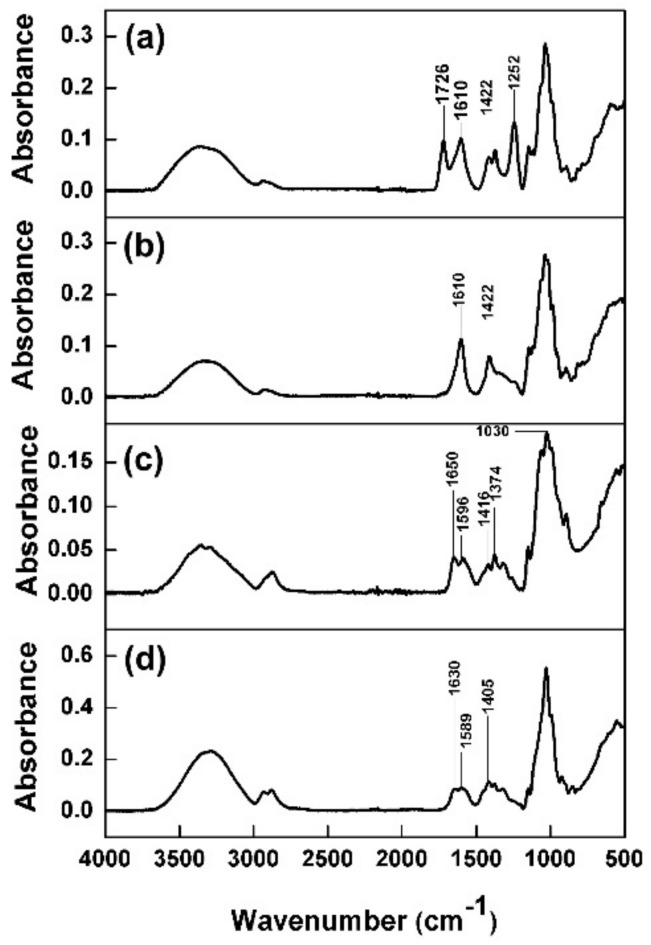
Attenuated total reflection-Fourier transform infrared (ATR-FTIR) spectra of (**a**) native Kg, (**b**) deacetylated Kg, (**c**) Ch, and (**d**) Kg-Ch sponge.

**Figure 3 polymers-13-00251-f003:**
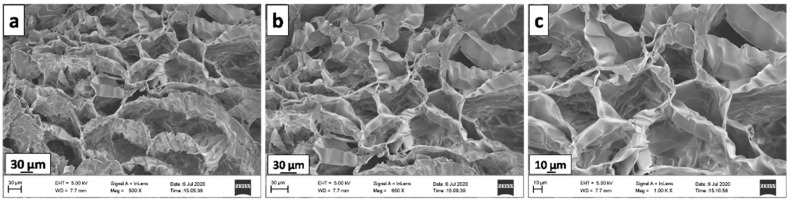
Scanning electron microscopy (SEM) images of Kg-Ch sponge at different magnifications. (**a**) 500X, scale 30 μm, (**b**) 650 X, scale at 30 μm, (**c**) 1000 X, scale 10 μm.

**Figure 4 polymers-13-00251-f004:**
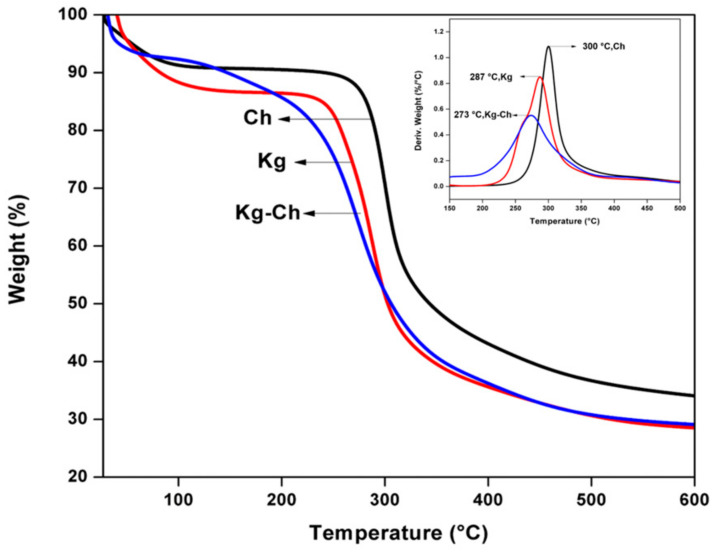
Thermograms showing weight percent as a function of temperature for Ch, Kg, and Kg-Ch sponge (derivative thermogravimetric curve insert).

**Figure 5 polymers-13-00251-f005:**
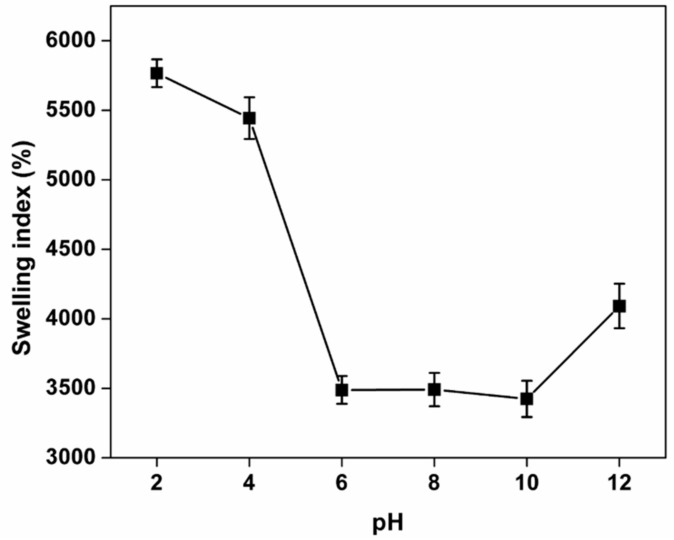
Effect of pH on the swelling index of Kg-Ch sponge.

**Figure 6 polymers-13-00251-f006:**
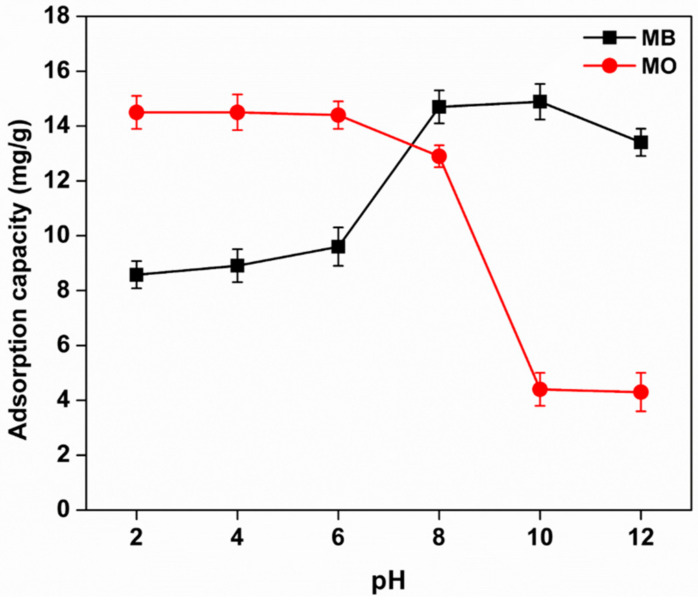
Effects of pH on the adsorption of methylene blue (MB) and methyl orange (MO) dyes onto the Kg-Ch sponge (*C*_0_ = 20 mg/L; contact time: 120 min; mass of Kg-Ch = 20 mg; temperature: 25 °C).

**Figure 7 polymers-13-00251-f007:**
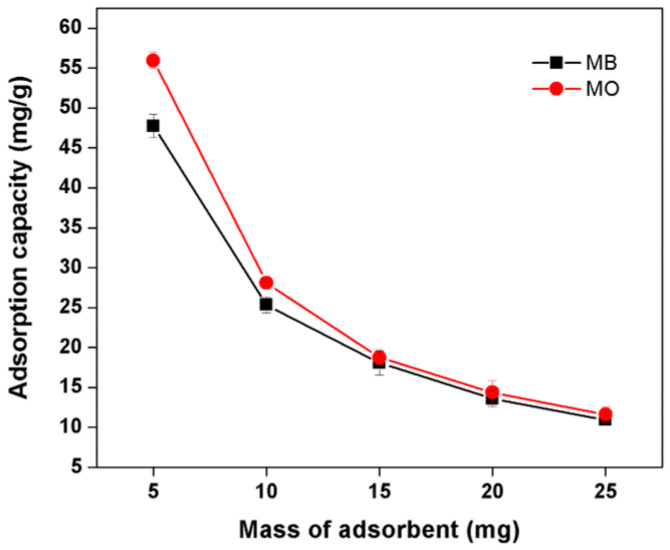
Effect of adsorbent dosage on the adsorption of MB and MO dyes onto the Kg-Ch sponge (*C*_0_ = 20 mg L^−1^; contact time: 120 min; pH = 8 for MB and 4 for MO; temperature: 25 °C).

**Figure 8 polymers-13-00251-f008:**
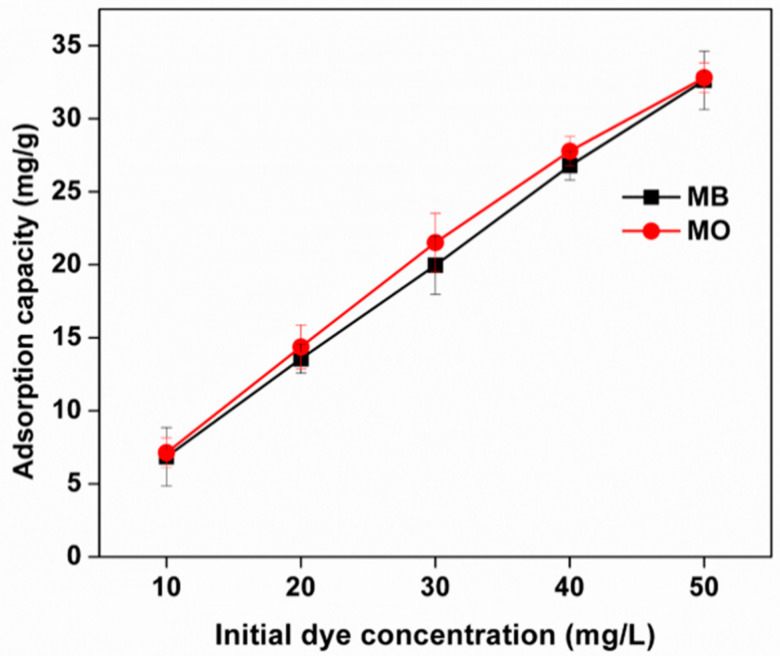
Effect of initial dye concentration on the adsorption of MB and MO dyes onto the Kg-Ch sponge (contact time: 120 min; pH of MB = 8.0 and MO = 4.0; mass of sponge = 20 mg; temperature: 25 °C).

**Figure 9 polymers-13-00251-f009:**
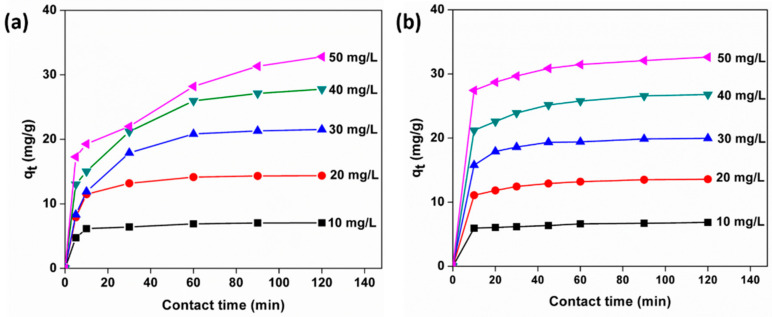
Effects of contact time on the adsorption capacities of (**a**) MO and (**b**) MB dyes onto the Kg-Ch sponge at different initial concentrations (*C_0_* = 10, 20, 30, 40, and 50 mg/L; temperature: 25 °C; contact time: 120 min).

**Figure 10 polymers-13-00251-f010:**
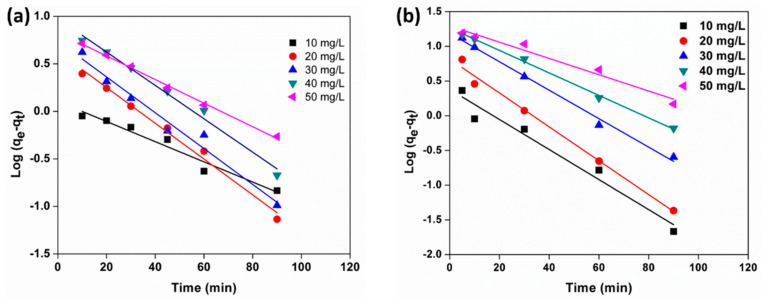
Pseudo-first-order model for the adsorption of (**a**) MB and (**b**) MO with different initial concentrations.

**Figure 11 polymers-13-00251-f011:**
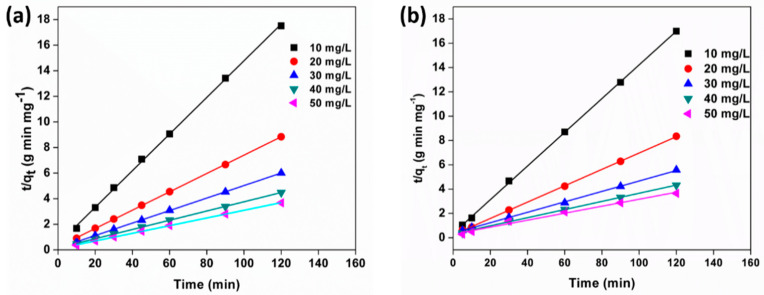
Pseudo-second-order model for the adsorption of (**a**) MB and (**b**) MO with different initial concentrations.

**Figure 12 polymers-13-00251-f012:**
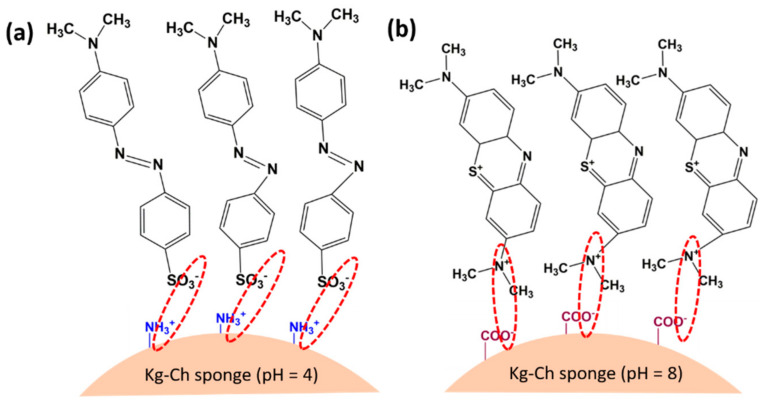
Schematic representation showing functional group interactions of (**a**) MO with Kg-Ch sponge at pH = 4.0 and (**b**) MB with Kg-Ch sponge at pH = 8.0.

**Figure 13 polymers-13-00251-f013:**
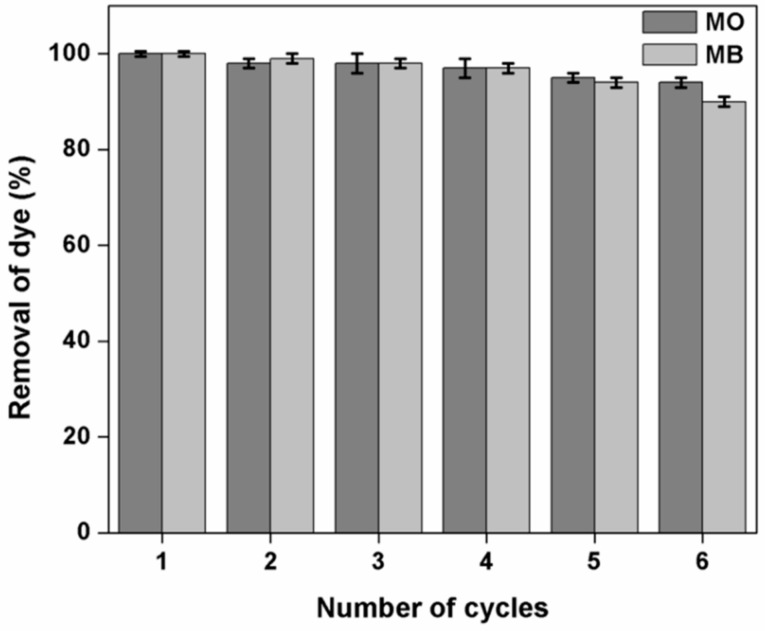
Reusability efficiency of the Kg-Ch sponge for the adsorption of MO and MB dyes (adsorbent dose = 20 mg; temperature = 30 °C and initial dye concentration = 20 mg L^−1^).

**Table 1 polymers-13-00251-t001:** Parameters of Langmuir and Freundlich models and R^2^ for MO and MB.

		Langmuir Model			Freundlich Model		
	q_e, exp_	q_0, cal_	K_L_	R^2^	K_F_	n	R^2^
MO	32.8	37.24	0.9	0.99	17.26	2.5	0.87
MB	32.6	80.58	0.1	0.91	7.89	1.27	0.99

**Table 2 polymers-13-00251-t002:** Pseudo-first-order kinetic parameters for MB and MO adsorption by Kg-Ch sponge.

		MB				MO		
*C*_0_ (mg/L)	q_e, exp_ ( mg/g)	q_e, cal_ (mg/g)	K_1_ (min^–1^)	R^2^	q_e, exp_ (mg/g)	q_e, cal_ (mg/g)	K_1_ (min^–1^)	R^2^
10	6.8	1.2	0.01	0.95	7.1	2.3	0.02	0.95
20	13.5	4.2	0.01	0.99	14.3	6.5	0.02	0.99
30	19.9	5.4	0.01	0.97	21.5	15.4	0.02	0.99
40	26.7	9.46	0.01	0.99	27.7	18.4	0.01	0.99
50	32.6	6.8	0.01	0.99	32.8	19.7	0.01	0.96

**Table 3 polymers-13-00251-t003:** Pseudo-second-order kinetic parameters for MB and MO adsorption by Kg-Ch sponge.

			MB			MO		
*C*_0_ (mg/L)	q_e, exp_ (mg/g)	q_e, cal_ (mg/g)	K_2_ (g mg^–1^ min^–1^)	R^2^	q_e, exp_ (mg/g)	q_e, cal_ (mg/g)	K_2_ (g mg^–1^ min^–1^)	R^2^
10	6.8	7.21	0.06	0.99	7.1	6.98	0.04	0.99
20	13.5	14.85	0.02	0.99	14.3	13.9	0.02	0.99
30	19.9	23.3	0.006	0.99	21.5	20.4	0.01	0.99
40	26.7	29.8	0.004	0.99	27.7	27.6	0.008	0.99
50	32.6	34.81	0.003	0.99	32.8	33.3	0.009	0.99

**Table 4 polymers-13-00251-t004:** Maximum dye (MB and MO) removal data: comparison of dye adsorption by Kg-Ch sponge with various other adsorbents.

Dye	Natural Adsorbents	q_e_ (mg/g)	Reference
MB	Polyvinyl alcohol—Xanthan gum hydrogel	4.16	[50]
	Guar gum-g-(acrylic acid-co-acrylamide-co-3-acrylamido propanoic acid)	27.06	[51]
	poly(acrylamide-co-*N*-methylacrylamide) grafted katira gum	21.45	[52]
	chitosan-magadiite hydrogel	45.25	[53]
	H_2_SO_4_ crosslinked magnetic chitosan	20.408	[54]
	Kg-Ch sponge	32.6	Present work
MO	γ-Fe_2_O_3_/chitosan composite films	29.41	[45]
	Xanthan gum was modified with acrylamide	29.56	[55]
	gellan gum-grafted-poly((2-dimethylamino) ethyl methacrylate)	25.8	[56]
	Karaya gum (KG) with 2-(methacryloyloxyethyl)trimethylammonium chloride	40.57	[57]
	Chitosan/diatomite composite	32.12	[58]
	Kg-Ch sponge	32.8	Present work

## Data Availability

Not applicable.

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
