# Peer review of "Eco-Friendly and Economic, Adsorptive Removal of Cationic and Anionic Dyes by Bio-Based Karaya Gum—Chitosan Sponge"

_polymers, 2021, doi:10.3390/polym13020251_

Round 1

Reviewer 1 Report

This manuscript reports the removal of dyes from aqueous solutions by their adsorption on a chitosan-based porous sponge. While there has been extensive research on dye sorption on natural materials, the present work has been conducted with care and the results are interesting. The manuscript is relatively well written but should be improved further prior to publication. The comments below are listed as guidance: 

1) Important information need to be included in the experimental method:

  • What was the heat rate for the thermogravimetric analysis?
  • Were the stability experiments conducted under constant shaking?
  • In the adsorption study, absorption spectroscopy was used to determine the solute concentration. What were the extinction coefficients (did the authors utilized a measured extinction coeff. or did they use a value reported in the literature)?
  • For the adsorption isotherms, non-linear regression methods should be preferred since linearization has be found to provide less representative results: Subramanyam et al International Journal of Environmental Science & Technology volume 6, 633–640 (2009).

2) Please add error bars in fig. 13 (regeneration of spent sorbents). In addition, the authors should provide information regarding the structural integrity and possible morphological changes in spent sorbents (i.e. after 6 regeneration cycles). 

3) The authors should compare their sorption capacity with other relevant studies reporting the uptake of dyes on natural sorbents (e.g. Z Yu et al. Nanomaterials 10 (1), 169, 2020 ; Shirazi et al International Journal of Industrial Chemistry volume 11, 101–110 (2020)).

4) The following references should be included in the section discussing the influence of solution pH and the possible formation of precipitates should be considered: Ren et al. Dalton Transactions42 (15), 5266-5274 (2013), Dichiara et al. ACS Appl. Mater. Interfaces, 7, 28, 15674–15680, 2015).

5) The porous sponge-like structure is quite desirable for continuous flow sorption processes (e.g. Goodman et al ACS Appl. Nano Mater. 1, 10, 5682–5690, 2018). Did the authors examine the potential of applying their material in fixed bed systems? 

Author Response

Authors' responses to reviewers' comment-  Please see the attachment

Reviewer 2 Report

The manuscript deals with "dye removal with using a y bio-based karaya gum - chitosan sponge". During recent years, industrialization and urbanization has led to a notable increase in wastewater discharge into the environment, mostly polluted with harmful organic contaminants (e.g., dyestuffs). Thus, the separation/elimination of these contaminants from water sources is a goal concern (https://doi.org/10.3390/w12082242). Using chitosan as a biopolymer, which is derived from chitin, has attracted researchers' attentions. The benefits of chitosan comprise its low cost, ease of polymerization and functionalization, and good stability (https://doi.org/10.3390/w11030551). 1. Page 1, Line 15; "...overall adsorption was controlled by pH..." Re-write it! For example, the adsorption capacity was decreased with increasing pH from 5 to 7. 2. Page 1, Line 17; "...adsorption isotherm of the anionic dye MO was found to correlate with Langmuir model..." Mention to the R2 values. 3. Page 1, Line 23; "the samples can be reused without loss of contaminant..." Should be corrected as "the samples could be reused without loss of contaminant..." 4. Page 1, Line 23; "...over successive adsorption- desorption cycles..." How many times? 5. How did you analysis dyes? Which method?

Author Response

Authors' responses to reviewers' comment- please see the attachment.

Round 2

Reviewer 2 Report

Reviewers’ comments have been addressed.